# Multimodal mismatch responses in mouse auditory cortex

**Magdalena Solyga[1], Georg B Keller[1,2]***

[1]Friedrich Miescher Institute for Biomedical Research, Basel, Switzerland; [2]Faculty of Science, University of Basel, Basel, Switzerland

## eLife Assessment

This well-designed study provides **important** findings concerning the way the brain encodes prediction about self-generated sensory inputs. The authors report that neurons in auditory cortex respond to mismatches in locomotion-driven auditory feedback and that those responses can be enhanced by concurrent mismatches in visual inputs. While there remain alternative explanations for some of the data, these findings provide **convincing** support for the role of predictive processing in cortical function by indicating that sensorimotor prediction errors in one modality influence the computation of prediction errors in another modality.

**\*For correspondence:**
georg.keller@fmi.ch

**Competing interest:** The authors declare that no competing interests exist.

**Abstract** Our movements result in predictable sensory feedback that is often multimodal. Based on deviations between predictions and actual sensory input, primary sensory areas of cortex have been shown to compute sensorimotor prediction errors. How prediction errors in one sensory modality influence the computation of prediction errors in another modality is still unclear. To investigate multimodal prediction errors in mouse auditory cortex, we used a virtual environment to experimentally couple running to both self-generated auditory and visual feedback. Using two-photon microscopy, we first characterized responses of layer 2/3 (L2/3) neurons to sounds, visual stimuli, and running onsets and found responses to all three stimuli. Probing responses evoked by audiomotor (AM) mismatches, we found that they closely resemble visuomotor (VM) mismatch responses in visual cortex (V1). Finally, testing for cross modal influence on AM mismatch responses by coupling both sound amplitude and visual flow speed to the speed of running, we found that AM mismatch responses were amplified when paired with concurrent VM mismatches. Our results demonstrate that multimodal and non-hierarchical interactions shape prediction error responses in cortical L2/3.

## Introduction

Neuronal responses consistent with prediction errors have been described in a variety of different cortical areas (*Audette and Schneider, 2023*; *Ayaz et al., 2019*; *Han and Helmchen, 2023*; *Heindorf et al., 2018*; *Keller et al., 2012*; *Liu and Kanold, 2022*) and across different species (*Eliades and Wang, 2008*; *Keller and Hahnloser, 2009*; *Stanley and Miall, 2007*). In V1, these responses are thought to be learned with experience (*Attinger et al., 2017*) and depend on local plasticity in cortex (*Widmer et al., 2022*). Prediction errors signal the unanticipated appearance or absence of a sensory input, and are thought to be computed as a deviation between top–down predictions and bottom–up sensory inputs (*Rao and Ballard, 1999*). One type of top–down signals that conveys a prediction of sensory input are motor-related signals (*Leinweber et al., 2017*). In auditory cortex (ACx), motor-related signals can modulate responses to self-generated vocalizations (*Eliades and Wang, 2008*) or to sounds coupled to locomotion (*Schneider et al., 2018*). ACx is thought to use these motor-related signals to compute audiomotor (AM) prediction error responses (*Audette et al., 2022*; *Audette and*

*Schneider, 2023*; *Eliades and Wang, 2008*; *Keller and Hahnloser, 2009*; *Liu and Kanold, 2022*). These AM prediction errors can be described in a hierarchical variant of predictive processing, in which top–down motor-related signals function as predictions of bottom–up sensory input. While parts of both auditory and visual processing streams are well described by a hierarchy, the cortical network as a whole does not easily map onto a hierarchical architecture, anatomically (*Markov et al., 2013*) or functionally (*St Yves et al., 2023*; *Suzuki et al., 2023*). One of the connections that does not neatly fit into a hierarchical model is the surprisingly dense reciprocal connection between ACx and V1 (*Clavagnier et al., 2004*; *Falchier et al., 2002*; *Ibrahim et al., 2016*; *Leinweber et al., 2017*; *Zhao et al., 2022*). From ACx to V1, this connection conveys a prediction of visual input given sound (*Garner and Keller, 2022*). What the reciprocal projection from V1 to ACx conveys is still unclear. In proposals for hierarchical implementations of predictive processing there are no such lateral connections, and there is no reason to assume prediction error computations in different modalities should directly interact at the level of primary sensory areas. Thus, we argued that the lateral interaction between V1 and ACx is a good starting point to investigate how non-hierarchical interactions are involved in the computation of prediction errors, and how multimodal interactions shape sensorimotor prediction errors.

Based on this idea, we designed an experiment in which we could couple and transiently decouple running speed in a virtual environment to both self-generated auditory feedback and self-generated visual flow feedback. While doing this, we recorded activity in L2/3 neurons of ACx using two-photon calcium imaging. Using this approach, we first confirmed that a substantial subset of L2/3 neurons in ACx responds to either auditory, visual (*Sharma et al., 2021*), or motor-related inputs (*Henschke et al., 2021*; *Morandell et al., 2023*; *Vivaldo et al., 2023*). While we found that L2/3 neurons in ACx responded to AM mismatches in a way that closely resembles visuomotor (VM) mismatch responses found in V1 (*Keller et al., 2012*), we found no evidence of responses to VM mismatch in ACx. However, when coupling both visual flow and auditory feedback to running, we found that L2/3 neurons in ACx non-linearly combine information about VM and AM mismatches. Overall, our results demonstrate that prediction errors can be potentiated by multimodal interactions in primary sensory cortices.

## Results
### Auditory, visual, and motor-related signals were intermixed in L2/3 of ACx

To investigate auditory, visual, and motor-related signals in mouse ACx, we combined an audiovisual virtual reality (VR) system with two-photon calcium imaging in L2/3 ACx neurons (*Figure 1A*). We used an adeno-associated viral (AAV) vector to express a genetically encoded calcium indicator (AAV2/1-EF1α-GCaMP6f-WPRE) in ACx (*Figure 1B–D*). Following recovery from surgery, mice were habituated to the VR setup (*Figure 1C*). We first mapped the location of the primary auditory cortex (A1) and the anterior auditory field (AAF) using widefield calcium imaging (*Figure 1C, E*). Based on these maps, we then chose recording locations for two-photon imaging of L2/3 neurons in either A1 or AAF. For the purposes of this work, we did not distinguish between A1 and AAF and will refer to these two areas here as ACx. To characterize basic sensory- and motor-related responses, we recorded neuronal responses to pure tones, full-field moving gratings, and running onsets. Throughout all experiments mice were free to run, and auditory and visual responses were pooled across both sitting and running conditions (unless explicitly stated otherwise). We first assessed population responses of L2/3 neurons evoked by sounds (pure tones presented at 4, 8, 16, or 32 kHz at either 60 or 75 dB sound pressure level [SPL]; *Figure 1F*) presented while the VR was off. While pure tones resulted in both increases and decreases in calcium activity in individual neurons (*Figure 1G*), the average population response exhibited a significant decrease in activity (*Figure 1H*). Next, we analyzed visual responses evoked by full-field drifting gratings (see Methods; *Figure 1I*). Visual stimulation resulted in a diverse response across the population of L2/3 neurons (*Figure 1J*) that was initially positive at the population level (*Figure 1K*). To quantify motor-related inputs, we analyzed activity during running onsets collected across all experimental conditions (*Figure 1L*). We found that the majority of neurons increased their activity during running onsets (*Figure 1M*), which was also reflected in a significant positive response on the population level (*Figure 1N*). Finally, we investigated how running modulates auditory and visual responses in ACx. In V1, running strongly increases responses to visual stimuli (*Niell and Stryker, 2010*), while in ACx running has been shown to modulate auditory responses in a variety of

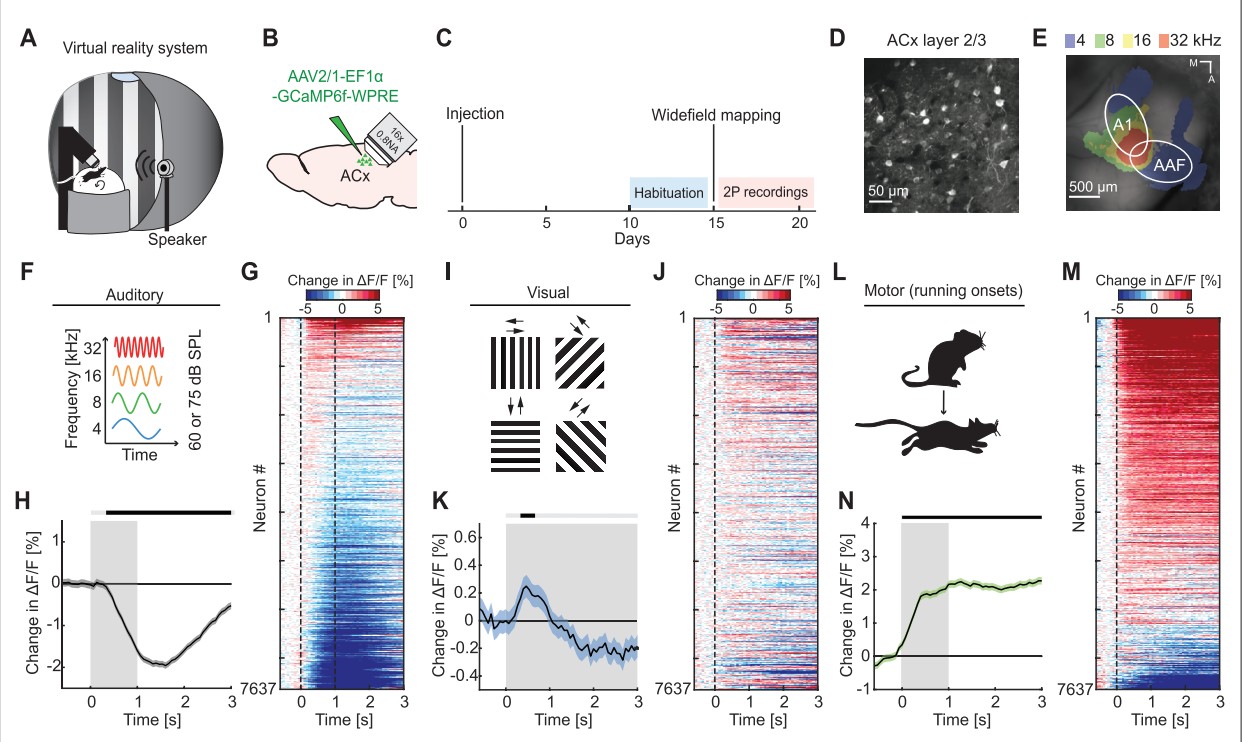

**Figure 1.** Auditory, visual, and motor-related signals were present in L2/3 of auditory cortex (ACx). (**A**) Schematic of the virtual reality system. For imaging experiments, mice were head-fixed and free to run on an air-supported spherical treadmill. For all recordings, the microscope was tilted 45° to the left to image left Acx. (**B**) Strategy for two-photon imaging of L2/3 ACx neurons. We injected an adeno-associated viral (AAV) vector to express a genetically encoded calcium indicator in ACx. (**C**) Timeline of the experiment. Starting 10 days after viral injection and window implantation surgery mice were habituated to the virtual reality setup without any visual or auditory stimulation for 5 days. We mapped ACx with widefield imaging to be able to target two-photon (2P) imaging to ACx. In one to six recording sessions, 1 session per day, we recorded from 7637 neurons in 17 mice. (**D**) Example two-photon image in L2/3 of Acx. (**E**) Example widefield mapping of ACx. Response maps reflect regions with the strongest response for each tested sound frequency. (**F**) The sound stimuli were 1 s long pure tones of 4, 8, 16, or 32 kHz played at 60 or 75 dB SPL, presented with randomized inter-stimulus intervals. (**G**) The average sound-evoked response of all L2/3 ACx neurons across all tested frequencies and sound levels. Sound is presented from 0 to 1 s. Red indicates an increase in activity, while blue indicates a decrease in activity. All responses are baseline subtracted. To avoid regression to the mean artifacts in plotting, the response heatmap is generated by splitting data in two halves by trials. The responses from the first half of trials are used to sort neurons by response strength and the average responses of the second half of trials are plotted for each neuron. To prevent graphical aliasing, the heatmaps are smoothed over 10 neurons for plotting. (**H**) The average sound-evoked population response of all ACx L2/3 neurons across all tested frequencies and sound levels (7637 neurons). Stimulus duration was 1 s (gray shading). Here and in subsequent panels, solid black lines represent mean and shading SEM. The horizontal bar above the plot marks time bins in which the response is statistically different from 0 (gray: not significant, black: p < 0.05; see Methods). See ***Supplementary file 1*** for all statistical information. (**I**) The visual stimuli we used were full-field drifting gratings of eight different directions, presented for 4–8 s with randomized inter-stimulus intervals. (**J**) As in (**G**), but for gratings onsets responses averaged across all orientations. (**K**) As in (**H**), but for the population response to grating onsets averaged across all orientations. (**L**) Motor-related activity was assessed based on responses upon running onsets. (**M**) As in (**G**), but for running onset responses. (**N**) As in (**H**), but for the average population response to running onsets. Only data from running onsets in which the mouse ran for at least 1 s (gray shading) were included.

The online version of this article includes the following figure supplement(s) for figure 1:

**Figure supplement 1.** Running exhibited differential effects on the responses to sound presentation and moving grating onsets.

different ways (***Audette et al., 2022***; ***Bigelow et al., 2019***; ***Henschke et al., 2021***; ***McGinley et al., 2015***; ***Morandell et al., 2023***; ***Schneider et al., 2014***; ***Vivaldo et al., 2023***; ***Yavorska and Wehr, 2021***; ***Zhou et al., 2014***). Separating auditory responses by running state, we found that sound-evoked responses of ACx neurons were overall similar during sitting and running, but exhibited a smaller decrease in activity when the mouse was sitting (***Figure 1—figure supplement 1A***). Visual responses consisted of an increase of activity during running, and a decrease of activity during sitting (***Figure 1—figure supplement 1B***). This is reminiscent of the running modulation effect observed on visual responses in V1 (***Niell and Stryker, 2010***). Thus, running appears to moderately and differentially modulate auditory and visual responses in L2/3 ACx neurons. Consistent with previous work,

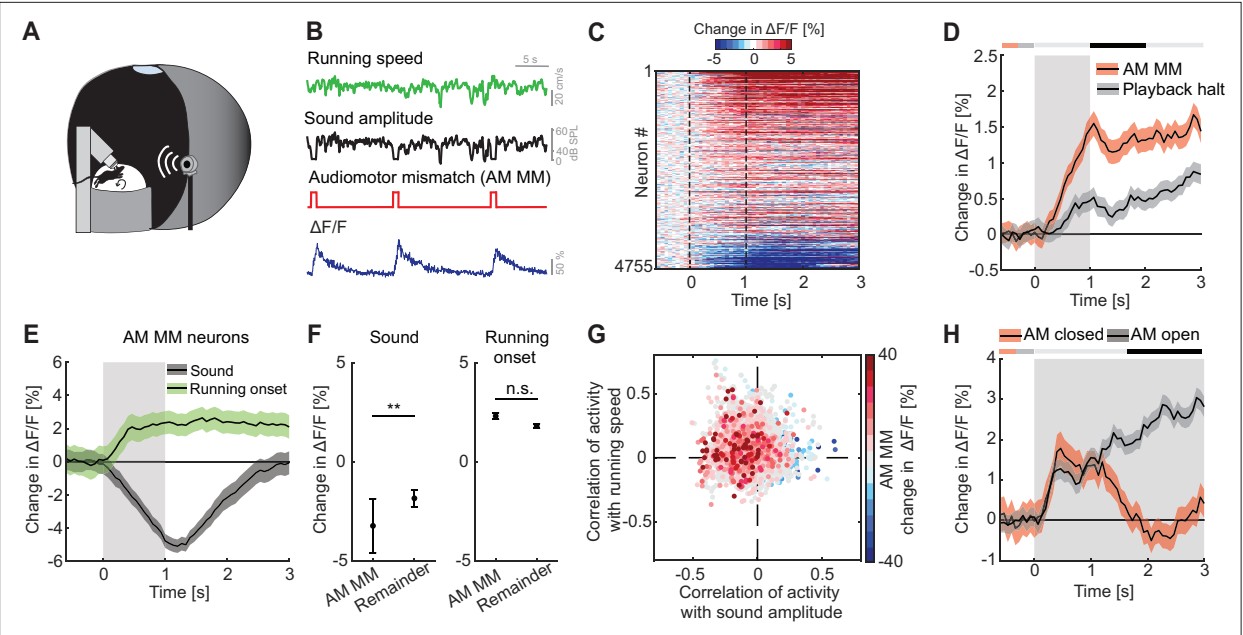

**Figure 2.** L2/3 neurons of auditory cortex (ACx) responded to audiomotor (AM) mismatch events. (**A**) Schematic of the virtual reality (VR) system used to study responses to AM mismatches. The sound amplitude of an 8-kHz pure tone was coupled to the running speed of the mouse on a spherical treadmill. These experiments were performed with the VR switched off. (**B**) In closed loop sessions, the running speed of the mouse was coupled to the sound amplitude. AM mismatches were introduced by briefly setting the sound amplitude to 0 for 1 s. Below, the calcium response of an example neuron to AM mismatch events. (**C**) Responses of all L2/3 ACx neurons to AM mismatches. The response heatmap is generated as described in *Figure 1G*. (**D**) The average population response of all L2/3 neurons to AM mismatches and sound playback halts (4755 neurons). AM mismatch duration was 1 s (gray shading). The horizontal bar above the plot marks time bins in which the AM mismatch response is statistically different from the playback halt response (gray: not significant, black: p < 0.05; see Methods). Here and in subsequent panels, solid black lines represent mean and shading SEM. (**E**) The average population response of AM mismatch neurons (5% of strongest responders) to sound stimulation (black) and running onsets (green). Same data as in *Figure 1H, N*, but subselected for AM MM neurons. Sound stimulation was 1 s (gray shading). (**F**) Comparison of the response strength of AM mismatch (MM) neurons to sound stimulation (left) and running onsets (right) compared to those of the remainder of the neuron population. Error bars indicate SEM. Here and elsewhere, n.s.: not significant; **p < 0.01. See *Supplementary file 1* for all statistical information. (**G**) Scatter plot of the correlations of calcium activity with sound amplitude (*x*-axis) and running speed (*y*-axis) in open loop sessions, for all neurons. The color-code reflects the strength of responses to AM mismatch in the closed loop session. Note, AM mismatch responsive neurons are enriched in the upper left quadrant. (**H**) The average population response to running onsets in closed (red) and open (black) loop sessions (only data from neurons for which we had at least two running onsets in both closed and open loop sessions are included here, see *Supplementary file 1*).

The online version of this article includes the following figure supplement(s) for figure 2:

**Figure supplement 1.** Controls for audiomotor (AM) mismatch responses.

**Figure supplement 2.** Opposing influence of sound and running on audiomotor (AM) mismatch neurons.

these results demonstrate that auditory, visual, and motor-related signals are all present in ACx and that running modulation influences L2/3 ACx neurons differently than in V1.

## L2/3 neurons in ACx responded to AM mismatch

To test whether auditory, visual, and motor-related signals are integrated in L2/3 neurons of ACx to compute prediction errors, we first probed for responses to AM mismatches. A mismatch in this context, is the absence of a sensory input that the brain predicts to receive from the environment, and thus a specific type of negative prediction error. We experimentally generated a coupling between movement and sensory feedback and then used movement as a proxy for what the mouse predicts to receive as sensory feedback. To do this with an auditory stimulus, we coupled the sound amplitude of an 8-kHz pure tone to the running speed of the mouse on the spherical treadmill such that sound amplitude was proportional to locomotion speed (*Figure 2A, B*). In this paradigm, a running speed of 0 corresponded to a sound amplitude of 0, while 30 cm/s running speed corresponded to a sound amplitude of 60 dB SPL. We refer to this type of session as closed loop. We then introduced AM mismatches by setting the sound amplitude to 0 for 1 s at random times (on

average every 15 s). An alternative approach to introduce AM mismatches would have been to clamp the sound amplitude to a constant value. However, based on an assumed analogy between sound amplitude and visual flow speed in VM mismatch paradigms, where we induce VM mismatch by setting visual flow speed to 0 (*Keller et al., 2012*), we chose the former. We found that AM mismatch resulted in a strong population response (*Figure 2C, D*). AM mismatch responses likely cannot be attributed to changes in running speed and do not result in a pupil response. We found no evidence of AM mismatch induced changes in running speed (*Figure 2—figure supplement 1A*) or pupil dilation (*Figure 2—figure supplement 1B*). Interestingly, the AM mismatch response was already apparent in the first closed loop session with AM coupling that the mice ever experienced, suggesting that this coupling is learned very rapidly (*Figure 2—figure supplement 1C*). To test whether AM mismatch responses can be explained by a sound offset response, we performed recordings in open loop sessions that consisted of a replay of the sound profile the mouse had self-generated in the preceding closed loop session. Mice were free to run during this session and did so at similar levels as during the closed loop session (*Figure 2—figure supplement 1D*). The average response to the playback of sound halt during the open loop session was significantly less strong than the average response to AM mismatch (*Figure 2D*, for detailed discussion on the relationship between mismatch responses and offset responses, see the authors' response to reviewers' recommendations). In contrast to visual playback halt responses in V1 (*Vasilevskaya et al., 2023*), we found no evidence of a running modulation of the response to the playback halt (*Figure 2—figure supplement 1E*). Thus, L2/3 neurons in ACx respond to AM mismatch in a way similar to how the L2/3 neurons in V1 respond to VM mismatch.

Assuming that AM mismatch responses are computed as a difference between an excitatory motor-related prediction and an auditory stimulus driven inhibition, we would expect the neurons with high AM mismatch responses to exhibit opposing influence of motor-related and auditory input. To test this, we selected the 5% of neurons with the strongest responses to AM mismatch and quantified the responses of these neurons to sound stimulation and running onsets (as for the sound and running onset responses shown in *Figure 1*). Consistent with a model of a subtractive computation of prediction errors, we found that AM mismatch neurons exhibited a strong reduction in activity in response to sound stimulation and an increase of activity on running onsets (*Figure 2E*). Given that mismatch responses are likely enriched in the superficial part of L2/3 (*O'Toole et al., 2023*), and that in our two-photon imaging experiments we also preferentially recorded from more superficial neurons, we suspect that our population is enriched for mismatch neurons. Consistent with this interpretation, we observed a strong population response to AM mismatches (*Figure 2C, D*) and a decrease in population activity in response to sound stimulation (*Figure 1H*). Nevertheless, sound-evoked responses were significantly more negative in neurons strongly responsive to AM mismatch, than for the remainder of the L2/3 neuronal population (*Figure 2F*). This effect was similar when we used different thresholds for the selection of AM mismatch neurons (10% or 20% of neurons with the strongest response to AM mismatch; *Figure 2—figure supplement 2*). Consistent with a sound driven reduction of activity and running related increase of activity in AM mismatch neurons, the correlation of calcium activity of AM mismatch neurons was predominantly negative with sound amplitude and positive with running speed in open loop sessions (*Figure 2G*). This again resembles the properties of VM mismatch neurons in V1 (*Attinger et al., 2017*). If AM mismatch responses are computed as a difference between a locomotion driven excitation and a sound driven inhibition, we could also expect to find a correlation between the strength of mismatch response and the strength of sound playback halt responses. Even in the absence of locomotion driven excitation, a relief from sound driven inhibition could trigger an increase in calcium activity. When comparing AM mismatch responses with playback sound halt responses for all neurons, we do indeed find a positive correlation between the two (*Figure 2—figure supplement 1F*). Finally, again assuming a locomotion driven excitation and a sound driven inhibition, we should find systematic differences in closed and open loop running onsets. In closed loop running onsets in which the sound was coupled to movement, we indeed observed only a transient running onset response (*Figure 2H*). In contrast, in the open loop condition, running onset responses were sustained, possibly reflecting motor-related input that is not cancelled out by sensory input. This analysis highlights the difference in input processing depending on whether the animal is in the coupled or uncoupled condition and is consistent with subtractive interactions between prediction and sensory signals. Overall, these results suggest that the implementation of sensorimotor prediction

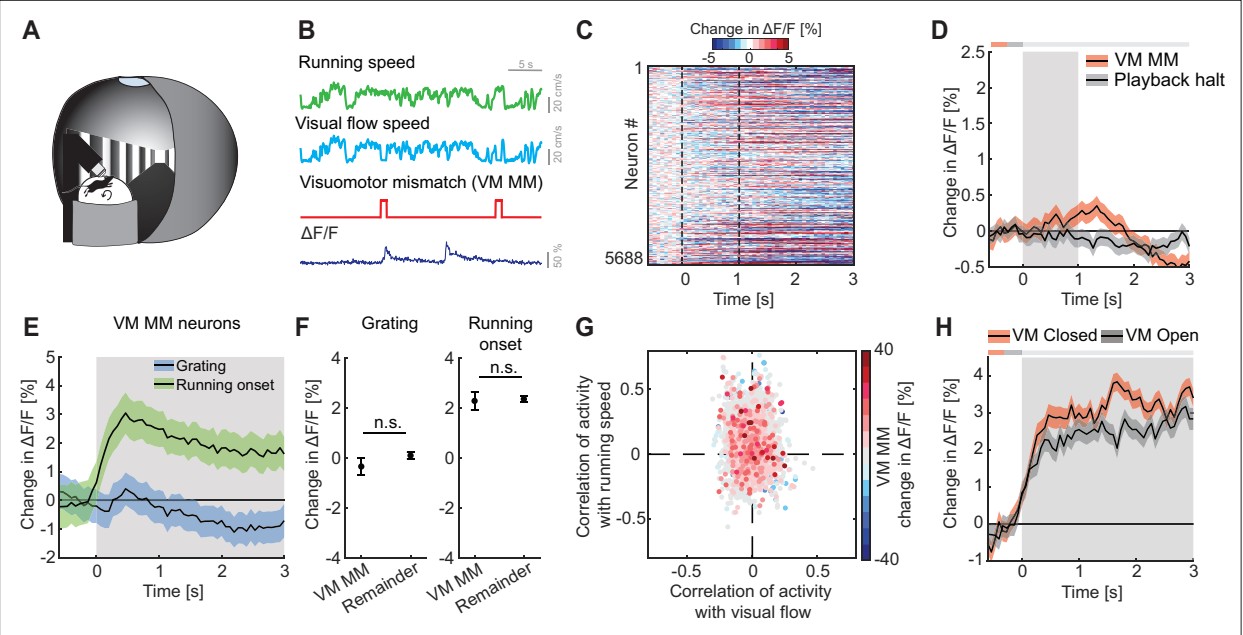

**Figure 3.** We found no evidence of visuomotor (VM) mismatch responses in auditory cortex (ACx). (**A**) Schematic of the virtual reality system used to measure VM mismatch responses. The visual flow of the virtual corridor was coupled to the running speed of the mouse on a spherical treadmill. There was no sound stimulus present in these experiments. (**B**) In closed loop sessions, the running speed of the mouse was coupled to the movement in a virtual corridor. VM mismatches were introduced by briefly setting visual flow speed to 0 for 1 s. Below, the calcium response of an example neuron to VM mismatch events. (**C**) Responses of all L2/3 ACx neurons to VM mismatches. The response heatmap is generated as described in *Figure 1G*. (**D**) The average population response of all L2/3 neurons to VM mismatches and visual flow playback halts (5688 neurons). Gray shading marks the duration of both stimuli. The horizontal bar above the plot marks time bins in which the VM mismatch response is statistically different from the playback halt response (gray: n.s., black: p < 0.05; see Methods). Here and in subsequent panels, solid black lines represent mean and shading SEM. (**E**) The average population response of VM mismatch neurons (5% of strongest responders) to grating stimulation (blue) and running onsets (green). Same data as in *Figure 1K, N*, but subselected for VM MM neurons. Stimulus duration was 4–8 s (gray shading). (**F**) Comparison of the response strength of VM mismatch (MM) neurons to visual stimulation (left) and running onsets (right) compared to those of the remainder of the neuron population. Error bars indicate SEM. Here and elsewhere, n.s.: not significant. See *Supplementary file 1* for all statistical information. (**G**) Scatter plot of the correlation of calcium activity with visual flow speed (*x*-axis) and running speed (*y*-axis) in open loop sessions for all neurons. The color-code reflects the strength of responses to VM mismatch in the closed loop session. Note, VM mismatch responsive neurons are scattered randomly. (**H**) The averaged population response to running onsets in closed (red) and open (black) loop session (only data from neurons for which we had at least two running onsets in both closed and open loop sessions are included here, see *Supplementary file 1*).

The online version of this article includes the following figure supplement(s) for figure 3:

**Figure supplement 1.** Controls for visuomotor (VM) mismatch responses.

error computation generalizes beyond V1 to other primary cortices and might be a canonical cortical computation in L2/3.

## We found no evidence of VM mismatch responses in L2/3 of ACx

VM mismatch responses are likely calculated in V1 (*Jordan and Keller, 2020*), and spread across dorsal cortex from there (*Heindorf and Keller, 2023*). To investigate multimodal mismatch responses, we first quantified the strength of these VM mismatch responses, which are independent of auditory input, in ACx. In these experiments, the running speed of the mouse was experimentally coupled to the visual flow speed in a virtual corridor, but not to sound feedback (*Figure 3A, B*). Note, there are several remaining sources of naturally occurring sound that increase with increasing running speed of the mouse and are still coupled (sound of mouse footsteps, sound of the treadmill rotating, etc.). However, these cannot be easily experimentally manipulated. We introduced VM mismatches by halting visual flow for 1 s at random times while the mice were running, as previously described (*Keller et al., 2012*; *Zmarz and Keller, 2016*). To control for visual responses independent of VM coupling, we used an open loop replay of the visual flow generated in the previous session (see Methods). We found that neither VM mismatches nor visual flow playback halts, which the mouse experienced in

open loop sessions, resulted in a measurable population response in ACx (*Figure 3C, D*). No significant changes in locomotion speed (*Figure 3—figure supplement 1A*) or pupil diameter (*Figure 3—figure supplement 1B*) were detected upon VM mismatch presentation. Selecting the 5% of neurons with the strongest responses to VM mismatches and quantifying their responses to grating presentations and running onsets, we found that these neurons exhibited positive responses to running onset and no significant response to grating stimuli (*Figure 3E*). These responses were not different from the population responses of the remainder of the neurons (*Figure 3F*). Quantifying the correlation of calcium activity with visual flow speed and running speed in the open loop session, we found that VM mismatch responsive neurons exhibited a distribution not different from chance (*Figure 3G*). Running onset responses were not different in closed and open loop conditions (*Figure 3H*). This suggests that motor-related predictions are not cancelled out by visual inputs in the ACx. We also found no evidence of a correlation between VM mismatch responses and playback halt responses (*Figure 3—figure supplement 1C*). Thus, while there may be a small subset of VM mismatch responsive neurons in L2/3 of ACx, we find no evidence of a VM mismatch response at the level of the L2/3 population.

## Mismatch responses were potentiated by multimodal interactions

Finally, we explored how multimodal coupling of both auditory and visual feedback to running speed influenced mismatch responses in L2/3 of ACx. To do this, we coupled both sound amplitude and visual flow speed to the running speed of the mouse in an audiovisual virtual environment (*Figure 4A, B*). We then introduced mismatch events by halting both sound and visual flow for 1 s to trigger a concurrent [AM + VM] mismatch (*Figure 4B*). The nomenclature here is such that the first letter in the pair denotes the sensory input that is being predicted, while the second letter denotes the putative predictor – the square brackets are used to denote that the two events happen concurrently. By putative predictor, we mean an information source available to the mouse that would, in principle, allow it to predict another input, given the current experimental environment. Thus, in the case of a [AM + VM] mismatch both the movement related prediction of visual flow and sound amplitude are violated. The [AM + VM] mismatch resulted in a significant response on the population level (*Figure 4C*). As with [AM] and [VM] mismatch, we found no evidence of a change in running speed or pupil diameter following [AM + VM] mismatch (*Figure 4—figure supplement 1A, B*). The concurrent experience of mismatch between multiple modalities could simply be the result of a linear combination of the responses to the different mismatch stimuli or could be the result of a non-linear combination. To test whether we find evidence of a non-linear combination of mismatch responses, we compared the [AM + VM] mismatch to [AM] and [VM] mismatch events presented alone. We found that the presentation of a [AM + VM] mismatch led to a significantly larger response than either an [AM] or a [VM] mismatch in isolation (*Figure 4D*). To test whether the linear summation of [AM] + [VM] mismatch responses could explain the response to the concurrent presentation [AM + VM], we compared the two directly, and found that the concurrent presentation [AM + VM] elicited a significantly larger response than the linear sum of [AM] + [VM] mismatch responses (*Figure 4E* and *Figure 4—figure supplement 1C*). Plotting the [AM + VM] mismatch responses against the linear sum of the [AM] + [VM] mismatch responses for each neuron, we found that while there is some correlation between the two, there is a subset of neurons (13.7%; red dots, *Figure 4F*) that selectively respond to the concurrent [AM + VM] mismatch, while a different subset of neurons (11.2%; yellow dots, *Figure 4F*) selectively responds to the mismatch responses in isolation. This demonstrates that mismatch responses in different modalities can interact non-linearly.

## Discussion

Consistent with previous reports, we found that auditory, visual, and motor-related signals are intermixed in the population of L2/3 neurons in ACx (*Figure 1H, K, N*). Responses to both motor-related (*McGinley et al., 2015*; *Schneider et al., 2014*; *Vivaldo et al., 2023*; *Yavorska and Wehr, 2021*; *Zhou et al., 2014*) and visual signals *Bigelow et al., 2022*; *Morrill and Hasenstaub, 2018*; *Sharma et al., 2021* have been reported across layers in ACx, with the strongest running modulation effect found in L2/3 (*Schneider et al., 2014*). Also, consistent with previous reports, we found that a subset of L2/3 neurons in ACx respond to AM prediction errors (*Figure 2C, D*; *Audette et al., 2022*; *Liu and Kanold, 2022*). In V1, it has been demonstrated that neurons signaling prediction errors exhibit

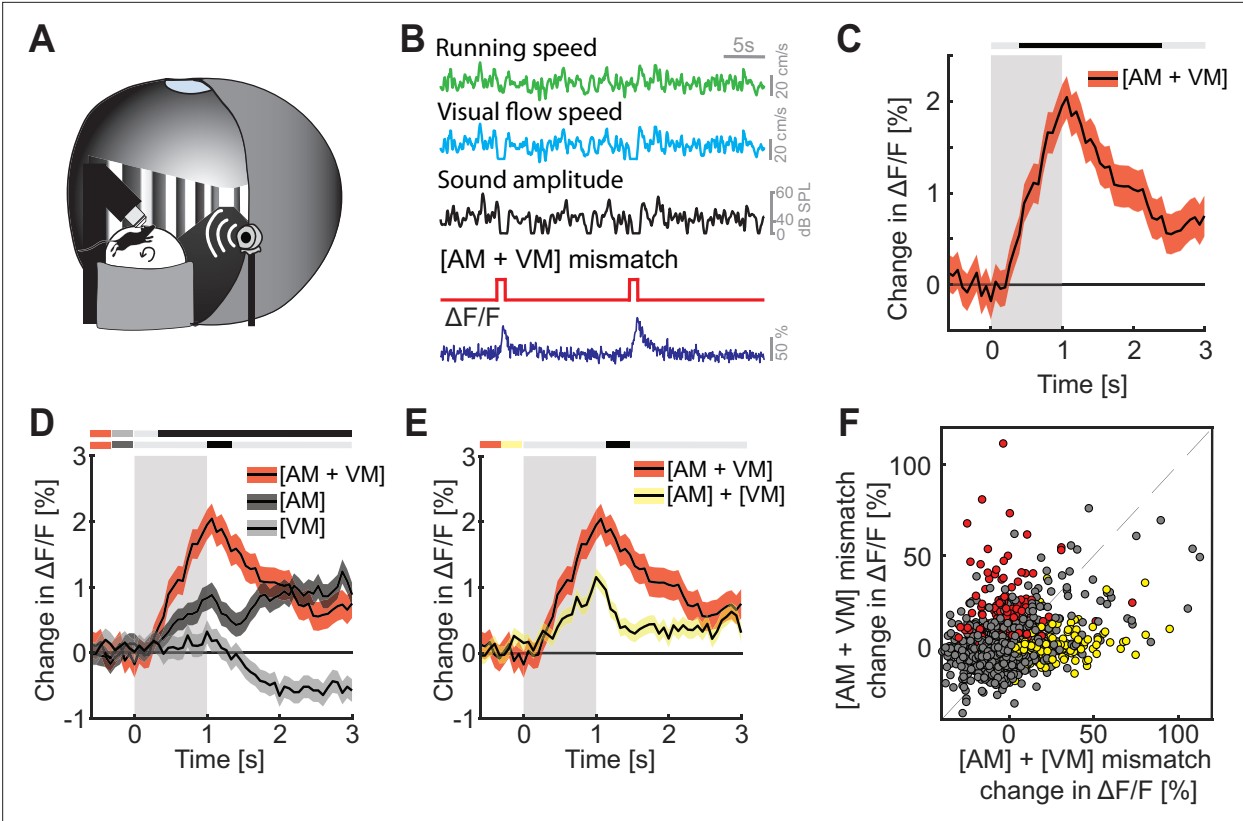

**Figure 4.** Mismatch responses were potentiated by multimodal interactions in L2/3 auditory cortex (ACx) neurons. (**A**) Schematic of the virtual reality system used to measure multimodal mismatch responses. Both the sound amplitude of an 8-kHz pure tone and the movement of the virtual corridor were coupled to the running speed of the mouse on a spherical treadmill. (**B**) Concurrent audio- and visuomotor [AM + VM] mismatches were introduced by simultaneously setting both sound amplitude and visual flow speed to 0 for 1 s. (**C**) The average population response of all L2/3 ACx neurons to concurrent [AM + VM] mismatches (3289 neurons). Gray shading marks the duration of the mismatch stimulus. The horizontal bar above the plot marks time bins in which the [AM + VM] mismatch response is statistically different from 0 (gray: n.s., black: p < 0.05; see Methods). Here and in subsequent panels, solid black lines represent mean and shading SEM. (**D**) Average population response of all L2/3 neurons to concurrent [AM + VM] mismatch and responses evoked by [AM] and [VM] mismatches presented in isolation (same data as shown in *Figures 2 and 3* but subselected to match the neurons that were recorded in both unimodal and multimodal mismatch paradigms; 3289 neurons). The horizontal bars above the plot mark time bins in which the [AM + VM] mismatch response is larger than the [AM] or the [VM] mismatch response (gray: n.s., black: p < 0.05; see Methods). The two short horizontal color bars to the left of time 0 indicate which two responses are being compared. Gray shading indicates the duration of the stimulus. See *Supplementary file 1* for all statistical information. (**E**) Average population response of all L2/3 neurons to a concurrent [AM + VM] mismatches compared to the linear sum of the responses evoked by [AM] and [VM] mismatches presented in isolation. Gray shading indicates the duration of the stimuli. (**F**) Scatter plot of the responses of all neurons to the concurrent [AM + VM] mismatches against the linear sum of the responses evoked by [AM] and [VM] mismatches presented in isolation. In red, the subset of neurons (13.7%) that exhibited selective responses to the concurrent [AM + VM] mismatch, and in yellow, the subset of neurons (11.2%) in which the linear sum of the responses to [AM] + [VM] mismatches presented in isolation was significant, while response to their concurrent presentation was not. Neurons without a significant response, or that are responsive to both, are shown in gray.

The online version of this article includes the following figure supplement(s) for figure 4:

**Figure supplement 1.** Controls for multimodal mismatch responses.

opposing influence of bottom–up visual and top–down motor-related inputs. This has been speculated to be the consequence of a subtractive computation of prediction errors (*Jordan and Keller, 2020*; *Keller et al., 2012*; *Leinweber et al., 2017*). Our findings now reveal a similar pattern of opposing influence in prediction error neurons in primary ACx that exhibit a positive correlation with motor-related input and a negative correlation with auditory input (*Figure 2G*). This would be consistent with the idea that both VM and AM prediction errors are computed as a subtractive difference between bottom–up and top–down inputs. Based on this, it is conceivable that this type of computation extends also beyond primary sensory areas of cortex and may be a more general computational principle implemented in L2/3 of cortex.

While there are many similarities between the AM mismatch responses we describe here in ACx and previously described VM mismatch responses in visual cortex (*Attinger et al., 2017*; *Keller et al., 2012*; *Widmer et al., 2022*; *Zmarz and Keller, 2016*), there are several notable differences. Specifically, AM responses in ACx (*Figure 2D*) appear to be more sustained than VM mismatch responses in visual cortex. We have no explanation for why this might be the case. Additionally, unlike the observed visual flow playback halt responses in V1 and their increase during running (*Vasilevskaya et al., 2023*), we found only very weak responses to sound playback halts and no evidence of an influence of running on these responses (*Figure 2—figure supplement 1E*). A decreased dependence of playback halt responses on running may be caused by the fact that the mouse generates additional sounds while running on an air-supported treadmill (e.g., treadmill rotation, changes in airflow, footsteps). These sounds correlate with running speed intensity and may reduce the relative salience of sound playback halts during running. Thus, differences in the experimental control of the sensorimotor coupling for a different sensory modalities could account for some of the observed differences between the AM and VM mismatch responses.

Finally, we found that concurrent prediction errors in multiple modalities result in an increase in prediction error response that exceeds a linear combination of the prediction error responses in single modalities (*Figure 4E*), with a subset of neurons selectively responding only to the combination of prediction error responses (*Figure 4F*). A similar non-linear relationship has been described between auditory and visual oddball responses in both ACx and V1 (*Shiramatsu et al., 2021*). At this point, it should be kept in mind that deviations from linearity in terms of spiking responses are difficult to assess using calcium imaging data. However, given that the difference between the concurrent presentation and the linear sum of the two individual mismatch responses was approximately a factor of two (*Figure 4E*), and the fact that we found a population of neurons that responds selectively to the concurrent presentation of both mismatches, we suspect that also the underlying spiking responses are non-linear. What are the mechanisms that could underlie this interaction? Neurons in ACx have access to information about VM mismatches from at least two sources. In widefield calcium imaging, VM mismatch responses are detectable across most of dorsal cortex (*Heindorf and Keller, 2023*). Thus, we speculate that VM mismatch responses are present in long-range cortico-cortical axons from V1, or possibly in L4, L5, or L6 neurons in ACx. Alternative sources of VM mismatch input are neuromodulatory signals. Locus coeruleus, for example, drives noradrenergic signals in response to VM mismatches across the entire dorsal cortex (*Jordan and Keller, 2023*). However, given that noradrenergic signals only weakly modulate responses in L2/3 neurons in V1 (*Jordan and Keller, 2023*), it is unclear if the broadcasted noradrenergic signals could non-linearly potentiate the AM mismatch responses of ACx neurons. We speculate that cholinergic signals are also unlikely to contribute to this effect. In V1 there are no cholinergic responses to VM mismatch (*Yogesh and Keller, 2023*). However, given that ACx and V1 receive cholinergic innervation from different sources (*Kim et al., 2016*), we cannot rule out the possibility that cholinergic signals in ACx respond to VM mismatch. In sum, given that the [AM + VM] mismatch responses do not simply appear to be an amplified variant of the [AM] mismatch responses (*Figure 4F*), we speculate that [AM + VM] mismatch responses are primarily driven by long-range cortico-cortical input from V1 that interacts with a local computation of [AM] mismatch responses in the L2/3 ACx circuit.

Lateral interactions in the computation of prediction errors between sensory streams are not accounted for by hierarchical variants of predictive processing. In these hierarchical variants, prediction errors are computed as a comparison between top–down and bottom–up inputs (*Rao and Ballard, 1999*). To explain the lateral interactions between prediction errors likely computed in ACx (AM mismatch responses) and prediction errors likely computed in V1 (VM mismatch responses) that we describe here, we will need new variants of predictive processing models that include lateral and non-hierarchical interactions. Thus, our results demonstrate that mismatch responses in different modalities interact non-linearly and can potentiate each other. The circuit mechanisms that underlie this form of multimodal integration of mismatch responses are still unclear and will require further investigation. However, we would argue that the relatively strong multimodal interaction demonstrates that unimodal and hierarchical variants of predictive processing are insufficient to explain cortical mismatch responses – if predictive processing aims to be a general theory of cortical function, we will need to explore non-hierarchical variants of predictive processing.

# Methods

## Key resources table

| Reagent type (species) or resource | Designation | Source or reference | Identifiers | Additional information |
|---|---|---|---|---|
| Strain, strain background (adeno-associated virus) | AAV2/1-EF1α-GCaMP6f-WPRE ($10^{14}$ GC/ml) | FMI vector core | vector.fmi.ch | |
| Chemical compound, drug | Fentanyl citrate | Actavis | CAS 990-73-8 | Anesthetic compound |
| Chemical compound, drug | Midazolam (Dormicum) | Roche | CAS 59467-96-8 | Anesthetic compound |
| Chemical compound, drug | Medetomidine (Domitor) | Orion Pharma | CAS 86347-14-0 | Anesthetic compound |
| Chemical compound, drug | Ropivacaine | Presenius Kabi | CAS 132112-35-7 | Analgesic compound |
| Chemical compound, drug | Lidocaine | Bichsel | CAS 137-58-6 | Analgesic compound |
| Chemical compound, drug | Buprenorphine | Reckitt Benckiser Healthcare | CAS 52485-79-7 | Analgesic compound |
| Chemical compound, drug | Ophthalmic gel (Humigel) | Virbac | N/A | Ophthalmic gel |
| Chemical compound, drug | Flumazenil (Anexate) | Roche | CAS 78755-81-4 | Anesthetic antagonist |
| Chemical compound, drug | Atipamezole (Antisedan) | Orion Pharma | CAS 104054-27-5 | Anesthetic antagonist |
| Chemical compound, drug | N-Butyl-2-cyanoacrylate | Braun | CAS 6606-65-1 | Histoacryl |
| Chemical compound, drug | Dental cement (Paladur) | Heraeus Kulzer | CAS 9066-86-8 | |
| Chemical compound, drug | Meloxicam (Metacam) | Boehringer Ingelheim | CAS 71125-39-8 | Analgesic compound |
| Strain, strain background (*Mus musculus*) | C57BL/6 | Charles River | RRID:IMSR_JAX:000664 | |
| Software, algorithm | MATLAB (2021b) | The MathWorks | RRID:SCR_001622 | Data analysis |
| Software, algorithm | LabVIEW | National Instruments | RRID:SCR_014325 | Hardware control |
| Software, algorithm | Python | https://www.python.org/ | RRID:SCR_008394 | Virtual reality |
| Software, algorithm | Panda3D | http://panda3d.org/ | N/A | Virtual reality |

## Mice and surgery

All animal procedures were approved by and carried out in accordance with guidelines of the Veterinary Department of the Canton Basel-Stadt, Switzerland. C57BL/6 female mice (Charles River), between the ages of 7 and 12 weeks were used in this study. For cranial window implantation, mice were anesthetized using a mixture of fentanyl (0.05 mg/kg), medetomidine (0.5 mg/kg), and midazolam (5 mg/kg). Analgesics were applied perioperatively. Lidocaine was injected subcutaneously into the scalp (10 mg/kg s.c.) prior to the surgery. Mice underwent a cranial window implantation surgery at an age of between 7 and 8 weeks. First, a custom-made titanium head-plate was attached to the skull (right hemisphere) with dental cement (Heraeus Kulzer). Next, a 3-mm craniotomy was made over left ACx (4.2–4.4 mm lateral from the midline and 2.6–2.8 mm posterior from bregma) followed by four to six injections of approximately 200 nl each of the AAV vector: AAV2/1-EF1α-GCaMP6f-WPRE ($10^{13-14}$ GC/ml). A circular glass cover slip was glued (Ultragel, Pattex) in place to seal the craniotomy. Metacam (5 mg/kg, s.c.) and buprenorphine (0.1 mg/kg s.c.) were injected intraperitoneally for 2 days after completion of the surgery. Mice were returned to their home cage and group housed for 10 days prior to the first experiments.

## VR environment

All recordings were done with mice head-fixed in a VR system, as described previously (*Leinweber et al., 2014*). Mice were free to run on an air-supported polystyrene ball. Three types of closed loop conditions were used for the experiments. The rotation of the spherical treadmill was either coupled (1) to the sound amplitude of an 8-kHz pure tone (AM coupling), while the animal was locomoting

in near-darkness (low ambient light in the experimental rooms, primarily from computer screens); (2) to the movement in a virtual corridor (VM coupling); or (3) to both the sound amplitude of an 8-kHz pure tone and the movement in a virtual corridor (audio–visuo–motor coupling). For AM coupling, we used the running speed of the mouse to control the SPL of an 8-kHz pure tone presented to the mouse through a loudspeaker (see section Auditory stimulation). This closed loop coupling was not instantaneous but exhibited a delay of 260 ± 60 ms (mean ± STD). For VM coupling, the running speed of the mouse was coupled to the visual flow speed in the virtual environment projected onto a toroidal screen surrounding the mouse using a Samsung SP-F10M projector synchronized to the turnaround times of the resonant scanner of the two-photon microscope. The delay in the VM closed loop coupling was 90 ± 10 ms (mean ± STD). From the point of view of the mouse, the screen covered a visual field of approximately 240° horizontally and 100° vertically. The virtual environment presented on the screen was a corridor tunnel with walls consisting of vertical sinusoidal gratings. In auditory experiments, the mouse generated additional sounds while running on an air-supported treadmill (e.g., treadmill rotation, changes in airflow, footsteps), which correlated with running speed intensity. Prior to the recording experiments, mice were habituated to the setup, without any coupling, in 1–2 hr long sessions for up to 5 days, until they displayed regular locomotion. On the first recording day, the mice experienced all three types of closed loop conditions (AM, VM, or combined coupling) in a random order. Closed loop sessions were followed by open loop sessions, in which rotation of the spherical treadmill was decoupled from both the sound amplitude and the movement in the virtual corridor. During these open loop sessions, we replayed the amplitude modulated sound or the visual flow recorded in the previous closed loop session.

## Auditory stimulation

Sounds were generated with a 16-bit digital-to-analog converter (PCI6738, National Instruments) using custom scripts written in LabVIEW (LabVIEW 2020, National Instruments) at 160-kHz sampling rate, amplified (SA1, Tucker Davis Technologies, FL, USA) and played through an MF1 speaker (Tucker Davis Technologies, FL, USA) positioned 10 cm from the mouse's right ear. Stimuli were calibrated with a wide-band ultrasonic acoustic sensor (Model 378C01, PCB Piezotronics, NY, USA). To study sound-evoked responses, we used 4, 8, 16, and 32 kHz pure tones played at 60 and 75 dB SPL (1-s duration, at a randomized inter-stimulus interval 4 ± 1 s, 10 repetitions, 1 ms on and off-ramp, in a randomized order). For AM coupling experiments, we used an 8-kHz pure tone with a sound amplitude that varied between 40 and 75 dB SPL.

## Visual stimulation

For visual stimulation, we used full-field sinusoidal drifting grating (0°, 45°, 90°, and 270°, moving in either direction) in a pseudo-random sequence, each presented for a duration of 6 ± 2 s, with between two and seven repetitions, with a randomized inter-stimulus interval of 4.5 ± 1.5 s during which a gray screen was displayed.

## Running onsets

Running onsets were defined as the running speed crossing a threshold of 3 cm/s, where the average speed in the previous 3 s was below 1.8 cm/s. To separate trials with AM mismatch, VM mismatch, auditory stimulus, and grating stimulus based on locomotion state into those running and those while sitting, we used threshold of 0.3 cm/s in a 1-s window preceding the stimulus onset.

## Widefield calcium imaging

To establish a reference tonotopic map of A1 and AAF (*Figure 1E*), we performed widefield fluorescence imaging experiments on a custom-built microscope consisting of objectives mounted face-to-face (Nikon 85 mm/f1.8 sample side, Nikon 50 mm/f1.4 sensor side), as previously described (*Heindorf and Keller, 2023*). Blue illumination was provided by a light-emitting diode (470 nm, Thorlabs) and passed through an excitation filter (SP490, Thorlabs). Green fluorescence emission was filtered with a 525/50 bandpass filter. Images were acquired at a frame rate of 100 Hz on a sCMOS camera (PCO edge 4.2). The raw images were cropped on-sensor, and the resulting data were saved to disk with custom-written software in LabVIEW (National Instruments).

## Two-photon imaging

Calcium imaging of L2/3 neurons in A1 and AAF was performed using a modified Thorlabs Bergamo II microscope with a 16×, 0.8 NA objective (Nikon N16XLWD-PF), as previously described (*Leinweber et al., 2014*). To record in left ACx, the microscope was tilted 45° to the left. The excitation light source was a tunable, femtosecond-pulsed laser (Insight, Spectra Physics or Chameleon, Coherent) tuned to 930 nm. The laser power was adjusted to 30 mW. A 12-kHz resonance scanner (Cambridge Technology) was used for line scanning, and we acquired 400 lines per frame. This resulted in a frame rate of 60 Hz at a resolution of 400 × 750 pixels. We used a piezo-electric linear actuator (Physik Instrumente, P-726) to record from imaging planes at four different cortical depths, separated by 15 μm. This reduced the effective frame rate per layer to 15 Hz. The emission light was bandpass filtered using a 525/50-nm filter (Semrock), and signals were detected with a photomultiplier (Hamamatsu, H7422), amplified (Femto, DHCPCA-100), digitized at 800 MHz (National Instruments, NI5772), and bandpass filtered at 80 MHz with a digital Fourier-transform filter on a field-programmable gate array (National Instruments, PXIe-7965). Recording locations were visually registered against the reference images acquired with widefield imaging previously using blood vessels patterns.

## Widefield image analysis

Off-line data processing and data analysis were done with custom-written MATLAB scripts. Slow drifts in the fluorescence signal were removed using 8th percentile filtering with a 62.5-s moving window, similar to what was used for two-photon imaging data (*Dombeck et al., 2007*). Activity was calculated as the $\Delta F/F_0$, where $F_0$ was the median fluorescence over the entire recording session. For stimulus responses, we use a response window of 0.2 to 1.2 s following stimulus onset and a baseline window of −1 to 0 s before stimulus onset. The pixels with the strongest response (top 3–5% of response distribution) were used to mark the tonotopic areas corresponding to the different stimuli.

## Two-photon image analysis

Calcium imaging data were processed as described previously. In brief, raw images were full-frame registered to correct for lateral brain motion. Neurons were selected manually based on mean and maximum fluorescence images. Average fluorescence per neuron over time was corrected for slow fluorescence drift using an 8th percentile filter and a 66-s (or 1000 frames) window (*Dombeck et al., 2007*; *Keller et al., 2012*; *Leinweber et al., 2014*) and divided by the median value over the entire trace to calculate $\Delta F/F_0$. All stimulus–response curves were baseline subtracted. The baseline subtraction window was −0.5 to 0 s before stimulus onset. For quantification of responses during different onset types (auditory, visual, running, and mismatch), $\Delta F/F$ was averaged over the response time window (0.5 to 2.5 s after stimulus onset) and baseline subtracted (mean activity in a window preceding stimulus onset, −0.5 to 0 s). Onsets which were not preceded by at least 2 s of baseline or not followed by at least 3 s of recording time, were excluded from the analysis. Sessions with less than two onsets were not included in the analysis. For analysis of responses to the combination of [AM + VM] (*Figure 4*, *Figure 4—figure supplement 1*), traces were aligned to the onset of the AM mismatch (due to the difference in lag of AM and VM coupling described above, VM mismatch preceded AM mismatch by approximately 170 ms). To quantify the difference in average calcium responses as a function of time, we used a hierarchical bootstrap test for every five frames of the calcium trace (333 ms) and marked comparisons where responses were different (p < 0.05). Mismatch responsive neurons were selected based on the absolute response strength over the response time window (0.5–2.5 s). To infer spikes from calcium signals (*Figure 4—figure supplement 1C*), we used CASCADE (*Rupprecht et al., 2021*). To test whether running speed or pupil size changed upon mismatch presentation, we compared the values at MM presentation (0.5 to 1 s) to the baseline values in the time window preceding MM presentation (−0.5 to 0 s).

## Statistical tests

All statistical information for the tests performed in this manuscript is provided in *Supplementary file 1*. We used hierarchical bootstrapping (*Saravanan et al., 2020*) for statistical testing to account for the nested structure of the data (multiple neurons from one imaging site). We first resampled the data with replacement at the level of imaging sites, followed by resampling at the level of neurons. We then computed the mean responses across the resampled population and repeated this process

10,000 times. The probability of one group being different from the other was calculated as a fraction of bootstrap sample means which violated the tested hypothesis.

## Acknowledgements

We thank Tingjia Lu for the production of viral vectors and all the members of the Keller lab for discussion and support. This project has received funding from the Swiss National Science Foundation (GBK), the Novartis Research Foundation (GBK), and the European Research Council (ERC) under the European Union's Horizon 2020 research and innovation programme (grant agreement No 865617) (GBK).

## Additional information

### Funding

| Funder | Grant reference number | Author |
|---|---|---|
| Horizon 2020 - Research and Innovation Framework Programme | 865617 | Georg B Keller |
| Novartis Stiftung für Medizinisch-Biologische Forschung | | Georg B Keller |
| Swiss National Science Foundation | | Georg B Keller |

The funders had no role in study design, data collection and interpretation, or the decision to submit the work for publication.

### Author contributions

Magdalena Solyga, Conceptualization, Data curation, Formal analysis, Investigation, Methodology, Software, Validation, Visualization, Writing – original draft, Writing – review and editing; Georg B Keller, Conceptualization, Funding acquisition, Project administration, Resources, Software, Supervision, Writing – original draft, Writing – review and editing

### Author ORCIDs

Magdalena Solyga ⓘ http://orcid.org/0000-0003-2969-2963
Georg B Keller ⓘ https://orcid.org/0000-0002-1401-0117

### Ethics

All animal procedures were approved by and carried out in accordance with guidelines of the Veterinary Department of the Canton Basel-Stadt, Switzerland. License 2573.

Reviewer #1 (Public review): https://doi.org/10.7554/eLife.95398.4.sa1
Reviewer #2 (Public review): https://doi.org/10.7554/eLife.95398.4.sa2
Reviewer #3 (Public review): https://doi.org/10.7554/eLife.95398.4.sa3
Author response https://doi.org/10.7554/eLife.95398.4.sa4

## Additional files

### Supplementary files
MDAR checklist

Supplementary file 1. All information on statistical tests used in this manuscript. We used hierarchical bootstrap (*Saravanan et al., 2020*) or a correlation coefficient for all comparisons.

## Data availability

All data generated and analyzed during this study, along with the code used for analysis, have been uploaded to Zenodo with the URL https://zenodo.org/records/14193759.

The following dataset was generated:

| Author(s) | Year | Dataset title | Dataset URL | Database and Identifier |
|---|---|---|---|---|
| Solyga M, Keller G | 2024 | Multimodal mismatch responses in mouse auditory cortex | https://doi.org/10.5281/zenodo.14193759 | Zenodo, 10.5281/zenodo.14193759 |

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
