## [Editor Report · eLife Assessment]

This well-designed study provides **important** findings concerning the way the brain encodes prediction about self-generated sensory inputs. The authors report that neurons in auditory cortex respond to mismatches in locomotion-driven auditory feedback and that those responses can be enhanced by concurrent mismatches in visual inputs. While there remain alternative explanations for some of the data, these findings provide **convincing** support for the role of predictive processing in cortical function by indicating that sensorimotor prediction errors in one modality influence the computation of prediction errors in another modality.

---

## [Referee Report · Reviewer #1 (Public review)]

Summary:

The manuscript presents a short report investigating mismatch responses in the auditory cortex, following previous studies focused on visual cortex. By correlating mouse locomotion speed with acoustic feedback levels, the authors demonstrate excitatory responses in a subset of neurons to halts in expected acoustic feedback. They show a lack of responses to mismatch in he visual modality. A subset of neurons show enhanced mismatch responses when both auditory and visual modalities are coupled to the animal's locomotion.

While the study is well-designed and addresses a timely question, several concerns exist regarding the quantification of animal behavior, potential alternative explanations for recorded signals, correlation between excitatory responses and animal velocity, discrepancies in reported values, and clarity regarding the identity of certain neurons.

Strengths:

(1) Well-designed study addressing a timely question in the field.

(2) Successful transition from previous work focused on visual cortex to auditory cortex, demonstrating generic principles in mismatch responses.

(3) Correlation between mouse locomotion speed and acoustic feedback levels provides evidence for prediction signal in the auditory cortex.

(4) Coupling of visual and auditory feedback show putative multimodal integration in auditory cortex.

Weaknesses:

(1) Unclear correlation between excitatory responses and animal velocity during halts, particularly in closed-loop versus playback conditions.

(2) Ambiguity regarding the identity of the [AM+VM] MM neurons.

---

## [Referee Report · Reviewer #2 (Public review)]

Using multimodal closed-loop behavior and activity monitoring in the neocortex, Solyga and Keller show that the auditory cortex computes the deviation of current sensory input from expectations. Interestingly, in addition, mismatch responses within the auditory stream are non-linearly influenced by concurrent sensorimotor error computations in the visual pathway. These results suggest that non-hierarchical interactions (lateral relational cross-talk) must be considered when analyzing cortical models based on predictive processing. In my opinion, this is a fundamental study that addresses the question of hierarchical vs. no-hierarchical interactions across neocortical areas. Overall, I find the experiments elegantly designed, and the results robust, providing compelling evidence for non-hierarchical interactions across neocortical areas, and more specifically of exchange of sensorimotor prediction error signals across modalities. The authors thoroughly addressed the concerns raised. In my opinion, this has substantially strengthened the manuscript, enabling much clearer interpretation of the results reported.

---

## [Referee Report · Reviewer #3 (Public review)]

This study explores sensory prediction errors in sensory cortex. It focuses on the question of how these signals are shaped by non-hierarchical interactions, specifically multimodal signals arising from same level cortical areas. The authors used 2-photon imaging of mouse auditory cortex in head-fixed mice that were presented with sounds and/or visual stimuli while moving on a ball. First, responses to pure tones, visual stimuli and movement onset were characterized. The authors then made the running speed of the mouse predictive of sound intensity and/or visual flow (closed loop). Mismatches were created through the interruption of sound and/or visual flow for 1 second, disrupting the expected sensory signal. As a control, sensory stimuli recorded during the close loop phase were presented again decoupled from the movement (open loop). The authors suggest that auditory responses to the unpredicted interruption of the sound, which affected neither running speed nor pupil size, reflect mismatch responses. That these mismatch responses were enhanced when the visual flow was congruently interrupted, indicates cross-modal influence of prediction error signals.

This study's strengths are the relevance of the question and the design of the experiment. The authors are experts in the techniques used. Responses to the interruption of the sound are similar in quality, if not quantity, in the predictive and the control situation, yet the contribution of sound offset sensitivity to the observed mismatch responses is not discussed.

---

## [Author Response]

The following is the authors’ response to the previous reviews.

**Reviewer #1:**
I am satisfied with all clarifications and additional analyses performed by the authors.The only concern I have is about changes in running after [AM+VM] mismatches.The authors reported that they "found no evidence of a change in running speed or pupil diameter following [AM + VM] mismatch (Figures S5A)" (line 197).Nevertheless, it seems that there is a clear increase in running speed for the [AM+VM] condition (S5A). Could this be more specifically quantified? I am concerned that part of the [AM+VM] could stem from this change in running behavior. Could one factor out the running contribution?

Please excuse, this was unintentionally omitted. We have added the quantification to Table S1 and included the results of the significance test in (Fig S2A, Fig S4A and Fig S5A). The increase in running speed upon MM presentation (0.5 – 1 s), compared to the baseline running speed in the time window preceding MM presentation (-0.5 – 0 s), was not significant in any of the tested conditions.

In the process of adding the statistics, we noticed an unfortunate inconsistency in our figures that relates to Figure S5A. The data shown in all other Figures is aligned to the onset of audiomotor mismatch. In Figure S5A, however, the data were aligned to the onset of the visuomotor mismatch. As there is a differential delay in the closed loop coupling of auditory and visual feedback of approximately 170 ms (as described in the methods), visuomotor mismatch onset is slightly before audiomotor mismatch onset. We have corrected this now in the manuscript but have done the statistical analysis for both old and new versions of the figure. In neither case do we find evidence of a running speed response.

The authors thoroughly addressed the concerns raised. In my opinion, this has substantially strengthened the manuscript, enabling much clearer interpretation of the results reported. I commend the authors for the response to review. Overall, I find the experiments elegantly designed, and the results robust, providing compelling evidence for non-hierarchical interactions across neocortical areas and more specifically for the exchange of sensorimotor prediction error signals across modalities.

We are happy to hear!

**Reviewer #2:**
The incorporation of the analysis of the animal's running speed and the pupil size upon sound interruption improves the interpretation of the data. The authors can now conclude that responses to the mismatch are not due to behavioral effects.The issue of the relationship between mismatch responses and offset responses remains uncommented. The auditory system is sensitive to transitions, also to silence. See the work of the Linden or the Barkat labs (including the work of the first author of this manuscript) on offset responses, and also that of the Mesgarani lab (Khalighinejad et al., 2019) on responses to transitions 'to clean' (Figure 1c) in human auditory cortex. Offset responses, as the first author knows well, are modulated by intensity and stimulus length (after adaptation?). That responses to the interruption of the sound are similar in quality, if not quantity, in the closed and open loop conditions suggest that offset response might modulate the mismatch response. A mismatch response that reflects a break in predictability would presumably be less modulated by the exact details of the sensory input than an offset response. Therefore, what is the relationship between the mismatch response and the mean sound amplitude prior to the sound interruption (for example during the preceding 1 second)? And between the mismatch response and the mean firing rate over the same period?Finally, how do visual stimuli modulate sound responses in the absence of a mismatch? Is the multimodal response potentiation specific to a mismatch?

There are probably two points important to clarify before answering the question – just to make sure there is no semantic misunderstanding.

(1) In the jargon of predictive processing, a prediction error is a deviation from a predictable relationship. This can be sensorimotor coupling (as in audio- and visuomotor mismatch), stimulus history (as in oddball, or sound offset responses), surround sensory input (as in endstopping response and center-surround effects in visual processing), etc. A sound offset perceived by an animal in an open loop condition is thus a negative prediction error based on stimulus history (this assumes the animal has no way to predict the time of offset – as is the case in our experiments). We are primarily interested in our work here in characterizing negative prediction errors that result from motor-related predictions – hence the comparison we use is unpredictable sound offset in closed-loop coupling vs. unpredictable sound offset in open-loop coupling. The first is a mixture of an audiomotor prediction error and a stimulus history prediction error. The second is just a stimulus history prediction error. Thus, we compare the two types of responses to isolate the component that can only be attributed to audiomotor prediction errors.

(2) Audiomotor mismatch responses can of course be explained in a large variety of ways. For example, one could consider a sound offset a sensory stimulus. One could further assume that locomotion increases sensory responses. If so, one could explain audiomotor mismatch responses as a locomotion related gain of a sensory offset response. However, we need to further postulate that this locomotion related gain is stimulus specific, as for sound onset responses there is no detectable difference between locomotion and sitting. Thus, we are left with a model that explains audiomotor mismatch responses as a “stimulus specific locomotion gain of sensory responses”. This is correct – it is just not very satisfying, has no computational basis, and makes no useful predictions (see e.g. https://pubmed.ncbi.nlm.nih.gov/36821437/ for an extended treatise of exactly this point for visuomotor mismatch responses).

That responses to the interruption of the sound are similar in quality, if not quantity, in the closed and open loop conditions suggest that offset response might modulate the mismatch response.

Conceptually both a “sound offset” and an “audiomotor mismatch” are negative prediction errors. Could one describe the effect we see as an audiomotor mismatch modulating a sound offset? Certainly. But if the reviewer means modulate in the sense of neuromodulatory – we are not aware of a neuromodulatory responses that would be fast enough (or be strong enough to have these effects – we have looked into ACh, NA, and Ser (unpublished – no MM response)). Alternatively, they could simply add linearly (as predictive processing would predict). Given that AM mismatch responses are likely computed in auditory cortex, we see no reason to speculate that anything more complicated is happening than a linear summation of different prediction error responses.

A mismatch response that reflects a break in predictability would presumably be less modulated by the exact details of the sensory input than an offset response. Therefore, what is the relationship between the mismatch response and the mean sound amplitude prior to the sound interruption (for example during the preceding 1 second)? And between the mismatch response and the mean firing rate over the same period?

The reviewer’s intuition here – that mismatch responses have a lower resolution than what one thinks of as sensory responses (or sound offset responses) – is probably not warranted. Experiments that quantify the resolution of mismatch responses are relatively data intense – and to the best of our knowledge this has only been done once in the visual system for visuomotor mismatch responses (Zmarz and Keller, 2016). Here we found that visuomotor mismatch responses exhibited matched spatial (in visual space) resolution to that of visual responses.

Regarding the suggested analyses: In a closed loop session, the sound amplitude preceding the mismatch is directly related to the running speed of the mouse. In visual cortex, the amplitude of visuomotor mismatch responses linearly scales with running speed (and consequently visual flow speed) prior to the mismatch – as predicted by predictive processing. See e.g. figure 4B in (Zmarz and Keller, 2016). We have tried this analysis for audiomotor mismatches in the previous round of reviews, but we fear we do not have sufficient data to address this question properly. If we look at how mismatch responses change as a function of locomotion speed (sound amplitude) across the entire population of neurons, we have no evidence of a systematic change (and the effects are highly variable as a function of speed bins we choose). However, just looking at the most audiomotor mismatch responsive neurons, we find a trend for increased responses with increasing running speed (Author response image 1). We analyzed the top 5% of cells that showed the strongest response to mismatch (MM) and divided the MM trials into three groups based on running speed: slow (10-20 cm/s), middle (20-30 cm/s), and fast (>30 cm/s). Given the fact that we have on average 14 mismatch events in total per neuron, the analysis when split by running speed is under-powered.

**Author response image 1. sa4fig1:** The average response of strongest AM MM responders to AM mismatches as a function of running speed (data are from 51 cells, 11 fields of view, 6 mice).

Regarding the relationship between mismatch response and firing rate prior to mismatch, we are not sure we understand the intuition. Does the reviewer mean, the average firing rate of the mismatch neuron? Or the population mean? The first is likely uninterpretable as it is bound to be confounded by regression to the mean type artefacts. But in either case, we would have no prediction of what to expect.